# Anisotropic magnon damping by zero-temperature quantum fluctuations in ferromagnetic CrGeTe₃

Lebing Chen[1], Chengjie Mao[2], Jae-Ho Chung [3✉], Matthew B. Stone [4], Alexander I. Kolesnikov [4], Xiaoping Wang [4], Naoki Murai[5], Bin Gao[1], Olivier Delaire [2✉] & Pengcheng Dai [1✉]

Spin and lattice are two fundamental degrees of freedom in a solid, and their fluctuations about the equilibrium values in a magnetic ordered crystalline lattice form quasiparticles termed magnons (spin waves) and phonons (lattice waves), respectively. In most materials with strong spin-lattice coupling (SLC), the interaction of spin and lattice induces energy gaps in the spin wave dispersion at the nominal intersections of magnon and phonon modes. Here we use neutron scattering to show that in the two-dimensional (2D) van der Waals honeycomb lattice ferromagnetic CrGeTe₃, spin waves propagating within the 2D plane exhibit an anomalous dispersion, damping, and breakdown of quasiparticle conservation, while magnons along the c axis behave as expected for a local moment ferromagnet. These results indicate the presence of dynamical SLC arising from the zero-temperature quantum fluctuations in CrGeTe₃, suggesting that the observed in-plane spin waves are mixed spin and lattice quasiparticles fundamentally different from pure magnons and phonons.

[1] Department of Physics and Astronomy, Rice University, Houston, TX 77005, USA. [2] Department of Mechanical Engineering and Materials Science, Duke University, Durham, NC 27708, USA. [3] Department of Physics, Korea University, Seoul 02841, Korea. [4] Neutron Scattering Division, Oak Ridge National Laboratory, Oak Ridge, TN 37831, USA. [5] J-PARC Center, Japan Atomic Energy Agency, Tokai, Ibaraki 319-1195, Japan. ✉email: jaehc@korea.ac.kr; olivier.delaire@duke.edu; pdai@rice.edu

In a magnetic ordered crystalline lattice, spin and lattice vibrations about their equilibrium positions form quasi-particles termed magnons (spin waves) and phonons (lattice waves), respectively[1]. Since these quasiparticles emerge from linearized theories that ignore all terms of order higher than quadratic and neglect interactions among the quasiparticles themselves, they are extremely stable against decay[2,3]. Moreover, because of the invariance of the ferromagnetic (FM) ground state under a spin rotation about the magnetization direction, the number of magnons is conserved[4] and spin waves have infinite lifetime throughout the Brillouin zone[5].

In a conventional local moment ferromagnet with a spin-rotational invariant Heisenberg Hamiltonian, spin waves are characterized by definite values of the $z$ projection of the total spin $S^z$, meaning every magnon has an intrinsic quantum number $|\Delta S^z| = 1$ and is conserved in magnon scattering processes[4]. In this case, magnon–magnon interactions play a minimum role in the modification of spin waves, as its Hamiltonian only contains the renormalization and the two-particle scattering term[4]. As a consequence, the intensity of spin waves measured by inelastic neutron scattering (INS) as a function of temperature in a local moment ferromagnet should only be renormalized by the Bose population factor[1]. Therefore, the energy width of peaks in spectra of spin waves with infinite lifetime should only be limited by the instrumental resolution, as seen in FM ordered $EuO_2$[5].

In systems where the spin and lattice degrees of freedom are coupled, the spin-lattice coupling (SLC) can modify spin waves in several ways. First, a static lattice distortion induced by SLC may affect the anisotropy of magnon exchange couplings, as seen in the spin waves of iron pnictides[6]. Second, time-dependent lattice vibrations interacting with spin waves may give rise to significant SLC. One possible consequence of such SLC is the formation of energy gaps in the spin-wave dispersion at the nominal inter-sections of magnon and phonon modes[7–9]. Alternatively, dynamic lattice deformation may change the spin coupling coefficient that may then decrease the lifetime of spin waves[10,11]. Nevertheless, the experimental observation of spin-wave damping in ferromagnets is rare[12–14], and may not arise from SLC[15].

Recently, SLC was suggested to be critical in understanding the ground state properties of two-dimensional (2D) van der Waals (vdW) FM $CrGeTe_3$ and $CrI_3$[16–21]. In these honeycomb ferromagnets, the superexchange coupling between nearest neighbor (NN) Cr–Cr bonds mediated with ligand Te/I atoms is FM, which competes with the antiferromagnetic (AF) Cr–Cr direct exchange, yielding a net FM interaction between NNs [Fig. 1(a, b)][22–26]. A consequence of this competition is the strong coupling between the inter-atomic distance and magnetic exchange couplings. For example, in $CrGeTe_3$, the AF Cr–Cr direct exchange decreases much faster with increasing Cr–Cr distance, compared with the Cr–Te–Cr superexchange [Fig. 1(b)]. According to ab-initio calculations[20,21], the FM exchange coupling has a slope of ~ 10 meV/Å as the inter-atomic distance between NN Cr–Cr pairs increases. Therefore, small dynamic lattice vibrations of the Cr atoms can directly affect spin waves in $CrGeTe_3$.

Experimentally, strong SLC has been suggested from Raman scattering, where optical phonon modes in $CrGeTe_3$ undergo narrowing in width and hardening in energy as the system is cooled below $T_C$[16]. However, Raman measurements can only probe a few optical phonon modes and certain magnon at the zone center $\Gamma$ point, and are unable to study spin waves throughout the Brillouin zone and their directional lifetime anisotropy. In addition, the in-plane lattice parameter $a$ of $CrGeTe_3$ displays negative thermal expansion around $T_C = 65\,K$[27], consistent with the calculations showing enhanced FM interaction with the expansion of the lattice. Finally, INS experiments found broadened spin waves in $CrGeTe_3$ throughout the Brillouin zone, suggestive of a strong SLC[26]. Although these results provided circumstantial evidence for SLC in $CrGeTe_3$, there is currently no direct experimental proof and a microscopic understanding of the SLC in $CrGeTe_3$ is lacking. Since $CrGeTe_3$ can be cleaved to a monolayer with long-range FM order[28] and potential for 2D spintronic devices due to possible dissipationless topological spin excitations[26], it is important to understand interactions of magnetic excitations with lattice vibrations because such SLC may fundamentally modify the topological nature of spin excitations.

In this work, we use INS to measure the spin and lattice dynamics in bulk single-crystal $CrGeTe_3$, complemented by

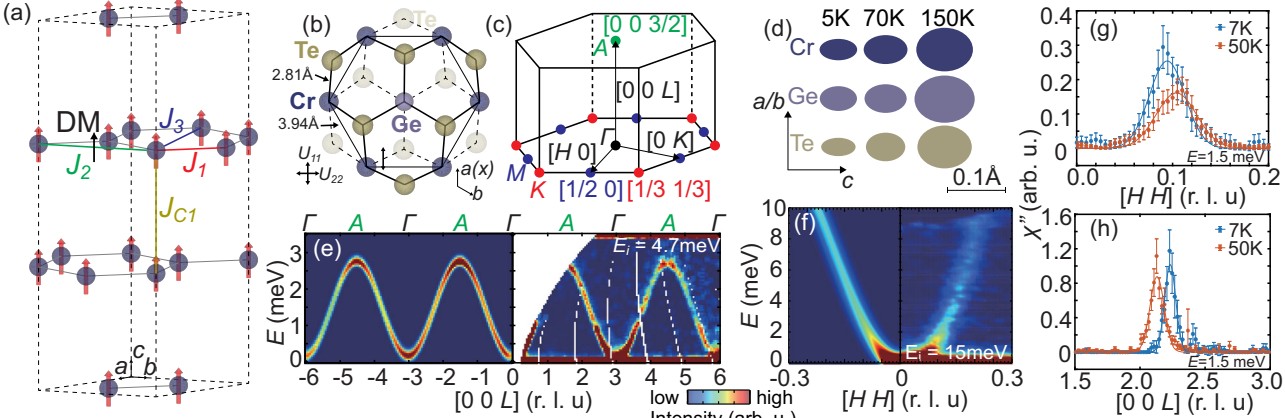

**Fig. 1 Real/reciprocal space of CrGeTe₃ and spin waves along the in-plane and c-axis directions. a** The hexagonal lattice of CrGeTe₃, showing only Cr³⁺ ions. The magnetic exchange couplings discussed in the main text are shown. **b** A closer view of the Cr (blue) hexagon together with the ligand Ge (gray) and Te (yellow) atoms. The bold (dashed) lines indicate bonds above (below) the Cr plane, and the Te atoms above (below) the Cr plane are drawn in heavy (light) yellow. The bode length of Cr–Te and Cr–Cr are specified. **c** The first Brillouin zone of the hexagonal lattice, high symmetry points Γ (black), M (blue), K (red), and A (green) are indicated. **d** A schematic picture of the temperature dependence of the root-mean square displacement of atoms in CrGeTe₃. **e** Spin waves along the [0, 0, L] direction. The color scale is applicable to all color plots in this paper. **f** The low-energy magnons along the [H, H, 3] direction. The left panels in **e**, **f** are the LSWT calculations convoluted with instrumental resolution using parameters given in the main text, and the right panels are experimental results. **g**, **h** Imaginary part of the dynamic susceptibility along the **g** [H, H] and **h** [0, 0, L] directions at different temperature. The error bars in **g**, **h** represent statistical errors of 1 standard deviation.

**Table 1 Neutron diffraction refinement results at 150, 70, and 5 K.**

| Atom/temperature | x | y | z | Occ. | $U_{11}$ | $U_{22}$ | $U_{33}$ | $U_{12}$ |
|---|---|---|---|---|---|---|---|---|
| Cr/150K | 0 | 0 | 0.3343 | 0.9899 | 0.00545 | 0.00545 | 0.00920 | 0.00273 |
| Cr/70K | 0 | 0 | 0.3345 | 0.9943 | 0.00240 | 0.00240 | 0.00550 | 0.00120 |
| Cr/5K | 0 | 0 | 0.3345 | 0.9750 | 0.00131 | 0.00131 | 0.00405 | 0.00066 |
| Ge/150K | 0 | 0 | 0.0581 | 1 | 0.00640 | 0.00640 | 0.01040 | 0.00320 |
| Ge/70K | 0 | 0 | 0.0582 | 1 | 0.00236 | 0.00236 | 0.00509 | 0.00118 |
| Ge/5K | 0 | 0 | 0.0584 | 1 | 0.00176 | 0.00176 | 0.00397 | 0.00088 |
| Te/150K | 0.6699 | 0.7043 | 0.2487 | 0.9504 | 0.00536 | 0.00528 | 0.00848 | 0.00271 |
| Te/70K | 0.6700 | 0.7045 | 0.2487 | 0.9907 | 0.00234 | 0.00235 | 0.00445 | 0.00117 |
| Te/5K | 0.6700 | 0.7044 | 0.2489 | 0.9716 | 0.00096 | 0.00100 | 0.00340 | 0.00044 |

density functional theory (DFT) calculations of phonon spectra and neutron diffraction measurements to determine temperature dependence of the atomic Debye-Waller factor. While spin waves along the c axis are resolution limited with well-defined dispersion following the expected behavior for a local moment ferromagnet, spin waves within the honeycomb lattice plane show broadening and damping throughout the Brillouin zone. Furthermore, the number of magnons is not conserved with increasing temperature. By comparing these results with DFT calculations of phonon spectra and neutron diffraction measurements of directional dependent atomic Debye-Waller factor, we conclude that the observed in-plane spin-wave anomalies arise from the large in-plane magnetic exchange coupling variations induced by anisotropic zero-temperature motion of Cr atoms. These results unveil the quantum zero-point motion induced SLC, suggesting that the observed in-plane spin waves are mixed spin and lattice quasiparticles fundamentally different from pure magnons and phonons.

## Results

**Real/reciprocal space and in-plane/c-axis spin waves.** Figure 1 (a, b) show real space images of CrGeTe$_3$, where the in-plane NN magnetic exchange $J_1$, second NN exchange $J_2$, third NN exchange $J_3$ and c-axis exchange $J_{c1}$ are marked. The corresponding reciprocal space with high symmetry points is shown in Fig. 1(c). Figure 1(d) summarizes a schematic picture of the temperature dependent root-mean square displacement of different atoms in CrGeTe$_3$ within the ab plane and along the c-axis direction determined from our single-crystal neutron diffraction analysis (Table 1) (see supplementary information for additional data and analysis). The left and right panels of Fig. 1(e) and (f) show calculated and measured spin-wave spectra along the c-axis and in-plane, respectively, where the calculation is obtained using a local moment Heisenberg Hamiltonian and linear spin-wave theory (LSWT)[1]. While the calculation and data agree rather well and are resolution limited throughout the Brillouin zone for spin waves along the c-axis [Fig. 1(e)], the measured spin waves within the ab-plane are weaker and broader than the calculated spectra [Figs. 1(f) and 2(a), (b)]. To test if population of magnons is conserved and follows the Bose population factor as expected for a local moment ferromagnet, we plot in Fig. 1(g) and 1(h) temperature dependence of the imaginary part of the dynamic susceptibility obtained using fluctuation-dissipation theorem, $\chi''(\mathbf{Q}, E) = (1 - e^{-E/k_B T})S(\mathbf{Q}, E)$, where $S(\mathbf{Q}, E)$ is the spin-wave intensity at energy $E$ and wave vector $\mathbf{Q}$ and $k_B$ is Boltzmann constant, at $E = 1.5$ meV in-plane and along the c-axis, respectively. Upon increasing temperature from 7 to 50 K in the FM ordered state, $\chi''(\mathbf{Q}, E)$ decreases for in-plane spin waves while it remains the same for magnons along the c-axis. This suggests a violation of magnon conservation for in-plane spin waves but not along the c-axis.

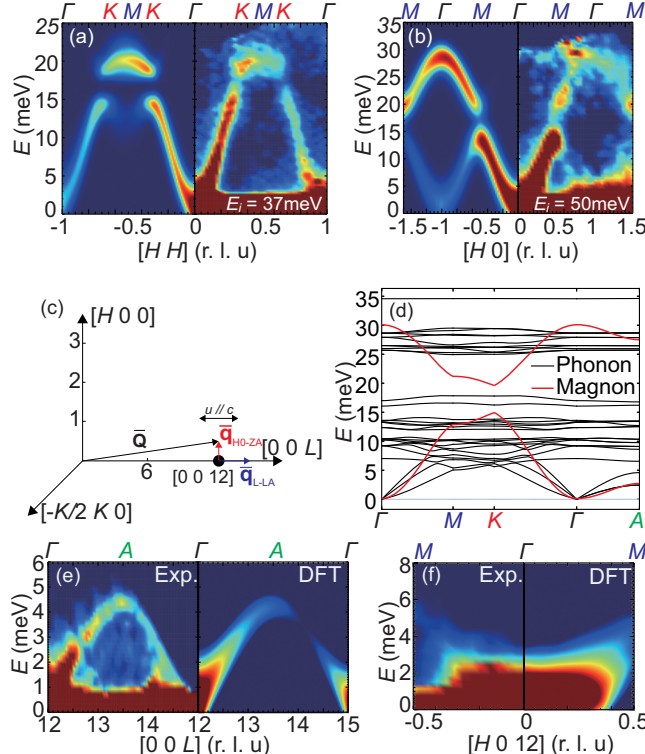

**Fig. 2 Magnon and phonon spectra of CrGeTe$_3$. a, b** INS spectra in the $[H, H]$ and $[H, 0]$ directions, respectively. **c** Experimental geometry for phonon modes probed in the experiment. **d** The calculated bulk phonon spectrum and fitted magnon spectrum. **e** The experimental (left) and DFT calculated (right) spectrum of $L(c)$ direction longitudinal acoustic (L-LA) phonon, respectively; **f** The experimental (left) and calculated (right) spectrum of in-plane direction transverse acoustic phonon, respectively.

The right panels in Fig. 2(a) and (b) show the overall spin-wave spectra in the $[H, K]$ plane, where we used incident neutron energies $E_i = 37$ and 50 meV, integrated over $L = [-5, 5]$, and $L = [-6, 6]$ to obtain the $[H, H]$ and $[H, 0]$ dispersion, respectively. Figure 2(c) shows the experimental geometry to probe phonons around wave vector $(0, 0, 12)$ where magnetic contributions can be safely ignored due to small magnetic form factor of Cr$^{3+}$ at this large $\mathbf{Q}$. Figure 2(d) compares the dispersions of the calculated magnons and phonons along the high symmetry directions within the ab plane. The $[0, 0, L]$ spin-wave spectrum along the c-axis in Fig. 1(e) is obtained using $E_i = 4.7$ meV. Inspection of Fig. 2(a) and (b) reveals clear acoustic and optical spin waves separated by a spin gap at the Dirac point, consistent with previous work[26] and very similar to spin waves in honeycomb lattice ferromagnets CrI$_3$[23–25] and CrBr$_3$[29,30].

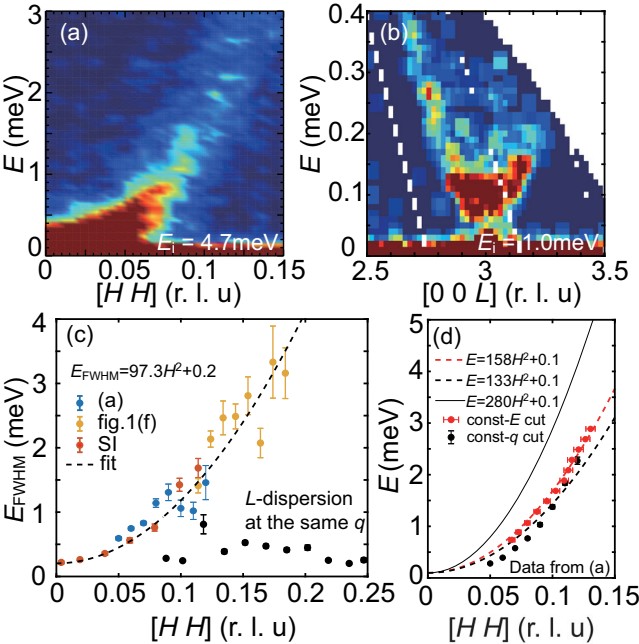

**Fig. 3 Details of in-plane magnon broadening. a** INS data along the $[H, H]$ direction with $E_i = 4.7$ meV and $L = 3$. **b** INS data along the $[0, 0, L]$ direction showing a ~ 0.1 meV spin gap at the $\Gamma$ point. **c** Magnon line-width (FWHM) as a function of reciprocal lattice vector $[H, H]$, and its comparison with the $L$-dispersion FWHM (black dots) at the same momentum transfer. **d** Long-wavelength magnon energy as a function of $[H, H]$ with quadratic fits (dashed lines). The solid line indicates the expected magnon dispersion calculated using LSWT with fitted parameters given in the text. The vertical and horizontal error bars in **c** and **d** represent uncertainty in fitted energy in meV and $[H, H]$ in r.l.u., respectively.

Consistent with earlier work[23–25], the magnon dispersion can be calculated with the Heisenberg Hamiltonian

$$H_0 = J_{ij}\mathbf{S}_i \cdot \mathbf{S}_j + \mathbf{A}_{ij} \cdot (\mathbf{S}_i \times \mathbf{S}_j) + D_z\left(S_i^z\right)^2 \quad (1)$$

where $J_{ij}$ includes the in-plane NN $J_1$, second NN $J_2$, third NN $J_3$ and $c$-axis exchange $J_{c1}$; $\mathbf{A}_{ij}$ is the the Dzyaloshinskii-Moriya (DM) term that exists only between second NN in honeycomb lattice according to Moriya's rule [Fig. 1(a) and (b)][31,32]. The single-ion anisotropy term $D_z = 0.033$ meV opens a 0.1 meV spin gap at the $\Gamma$ point [Fig. 3(a, b)]. By fitting the overall spin-wave dispersion, we find $J_1 = -2.76$ meV, $J_2 = -0.11$ meV, $J_3 = -0.33$ meV, $J_{c1} = -0.86$ meV, and $|\mathbf{A}| = 0.20$ meV. However, the calculated spin-wave spectra based on the Heisenberg model and instrumental resolution cannot fully explain the observed scattering intensity and broadening of the in-plane dispersion, especially for the optical magnons located at the $\Gamma$ point [Figs. 1(f) and 2(a, b)]. In contrast, the $c$-axis $[0, 0, L]$ dispersion agrees well with the Heisenberg Hamiltonian and spin waves are instrumental resolution limited [Fig. 1(e)]. Furthermore, we rule out the trivial broadening due to finite mosaic spreads of the co-aligned crystals (see supplementary information for additional data and analysis).

As a FM insulator, CrGeTe$_3$ has an electronic band gap of ~ 380 meV[33], which is orders of magnitude higher than the thermal and magnon energy. The observed magnon broadening and damping thus cannot be a result of magnon-magnon couplings from itinerant electrons[4]. Even considering magnon–magnon interaction, quasiparticle numbers in a ferromagnet should still be conserved, meaning that $\chi''(\mathbf{Q}, E)$, $E$ should

be independent of temperature in the FM state. Fig. 1(g) and (h) show the $\chi''(\mathbf{Q}, E)$ calculated from experimentally measured spin waves. While $\chi''(\mathbf{Q}, E)$ is highly temperature dependent along the in-plane direction, it is temperature independent along the $c$-axis. These results indicate the breaking of in-plane magnon number conservation, and can only be explained by anisotropic SLC.

**Low-energy acoustic phonons and calculated phonon dispersions.** To understand how phonons are coupled with spin waves, we use INS to measure acoustic phonon modes with polarization along the $c$-axis. Using Bragg peak position $(0, 0, 12)$ as the zone center $\Gamma$ point, we avoid strong magnetic scattering and can therefore directly probe lattice vibrations [Fig. 2(c)]. Experimentally, the probed phonon vibration direction is parallel to the momentum transfer $\mathbf{Q}$, and the phonon energy is a function of its reduced momentum vector $\mathbf{q}$ in the first Brillouin zone, where $\mathbf{q} = \mathbf{Q} - \mathbf{G}$ with $\mathbf{G}$ being a reciprocal lattice vector. Since any $\mathbf{q}$ in the first Brillouin zone will be an order of magnitude smaller than $\mathbf{G} = (0, 0, 12)$, we can safely assume the probed phonon spectrum around $(0, 0, 12)$ has a vibration direction along the $c(z)$-axis. To compare with INS experiments, we also use DFT to calculate the phonon dispersions and intensities in bulk CrGeTe$_3$ [Fig. 2(d)]. Although phonon and magnon spectra have similar bandwidth as well as many mutual crossover points as seen in Fig. 2(d), the optical spin waves from $M$ to $K$ point have no overlap with any phonon modes. Therefore, the optical magnon broadening observed in Fig. 2(a) and (b) cannot be a result of magnon-phonon coupling induced by mode crossovers[9,13]. The left and right panels of Fig. 2(e) show dispersions of the measured and calculated acoustic phonon along the $c$-axis, respectively. Overall, the measured phonon spectra is compatible to our ab-initio calculations and the reported phonon structure in 2D CrGeTe$_3$[20].

As discussed earlier, SLC can have several effects on the magnon and phonon spectra. First, a lattice distortion induced by the magnetic order can affect the exchange coupling as well as phonon energy, consistent with the observed phonon modes hardening below $T_C$ in CrGeTe$_3$[16]; Second, if the phonon and magnon modes intersect with each other, it will either open up a gap or broaden the magnon signal due to SLC. However, we find no sudden magnon broadening or energy gap at possible magnon-phonon cross points [Figs. 1(f), 2(a, b, d)]. Instead, we find two spin-wave anomalies: 1. The width of acoustic spin waves increases quadratically with increasing $\mathbf{q}$ [Fig. 3(a) and (c)]; 2. The low-energy spin-wave dispersion deviates from the calculated dispersion using Heisenberg Hamiltonian fits from the overall spin-wave spectra [Fig. 3(b) and (d)]. Figure 3(a) shows the low-energy spin-wave spectrum along the in-plane $[H, H]$ direction (see supplementary information for additional data and analysis). By carrying constant-$E$ and $\mathbf{Q}$ cuts from the spin-wave spectrum, we obtain $H$ dependence of the full width of half maximum (FWHM) of the magnon energy (lifetime). We subtracted the fitted FWHM by the calculated instrumental resolution and find $E_{FWHM} = \gamma H^2 + E_0$, where $E_0 = 0.2$ meV accounts for the additional broadening effect other than instrumental resolution (see Sec. 1.3 in the supplemental information for details) [Fig. 3(c)]. From the Heisenberg Hamiltonian fits to the overall spin-wave spectra in Fig. 2(a) and (b), we expect the spin-wave stiffness in the long-wavelength limit (small $\mathbf{q}$) to be $D = 4\pi^2 S(J_1 + 6J_2 + 4J_3) = 280.4$ meV/(r.l.u.)$^2$, clearly different from the observation of $D \approx 140$ meV/(r.l.u.)$^2$ [Fig. 3(d)]. For comparison, spin waves along the $c$-axis are resolution limited and can entirely be described by a Heisenberg Hamiltonian [Fig. 1(e)]. Therefore, our results suggest the presence of significant in-plane magnetic exchange coupling variations that cannot be accounted for by the average exchange

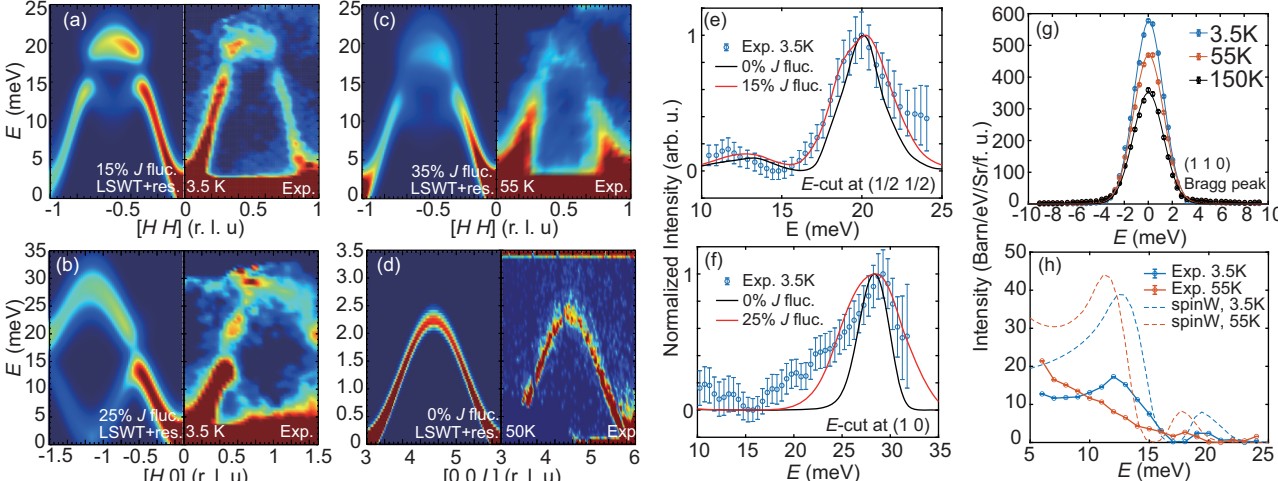

**Fig. 4 Simulations on magnon spectrum broadening and the total moment sum rule. a** LSWT calculations along the [H, H] direction with instrumental resolution, with 15% in-plane J fluctuation compared with experimental results from SEQUOIA at 3.5 K; **b** LSWT calculation along the [H, 0] direction with instrumental resolution, with 25% in-plane J fluctuation, compared with experimental results; **c** same calculation as in **a** with 35% in-plane J fluctuation and experimental results at 55 K; **d** The [0, 0, L] dispersion at 50 K, with no fluctuation of J in the calculation. **e, f** Respective E-cuts at the M point and Γ point for the experimental data (blue dots), calculation without J fluctuation (black lines) and with J fluctuation (red lines). **g** The (1, 1, 0) elastic peak at 3.5, 55, and 150 K, normalized to absolute units; **h** The INS intensity S(E) at 3.5 K and 55 K, averaged over the first Brillouin zone. The dashed lines are calculations of spin-wave intensities from the LSWT. The vertical error bars in **e** and **f** represent statistical errors of 1 standard deviation.

couplings obtained from a Heisenberg Hamiltonian fit to the overall spin-wave dispersion [Fig. 2(a) and (b)].

## Discussion

To quantitatively understand these observations, we consider the effect of SLC in a Heisenberg Hamiltonian

$$H_{SLC} = \sum_{ij} (\mathbf{S}_i \cdot \mathbf{S}_j) \frac{dJ_{ij}}{d\mathbf{u}} \mathbf{u}, \qquad (2)$$

where $\mathbf{u}$ is the dynamical lattice displacement of Cr atoms away from their equilibrium positions. Therefore, the total spin Hamiltonian will be modified as $H = H_0 + H_{SLC}$. Various DFT calculations estimated the SLC coefficient $dJ_{ij}/d\mathbf{u}$ in CrGeTe₃[20,21]. For example, the calculated NN exchange $J_1$ has an unidirectional relationship with atomic displacement $dJ_1/du_x^{Cr} = -8.48$ meV/Å, with x along the nearest Cr–Cr bond [Fig. 1(b)][21]. Figure 1(d) visualizes the temperature dependence of the in-plane ($U_{11}$) and c-axis ($U_{33}$) Debye-Waller factor in root-mean-square displacement for Cr, Ge, and Te obtained by crystal structure analysis of neutron single-crystal diffraction data of CrGeTe₃ (see supplementary information for additional data and analysis). By multiplying the calculated SLC coefficient and refined Debye-Waller factor, we can quantitatively estimate the fluctuation of the exchange interactions at different temperatures. At 5 K, the Cr in-plane $u_x^{Cr}$, as estimated by $u_x^{Cr} = \sqrt{U_{11}^{Cr}}$, equals to 0.036 Å and therefore can modulate the NN exchange $J_1$ by ~ 0.72 meV (FWHM = 2.36 × 0.036 × 8.48), including atomic displacements of Ge and Te atoms will have additional effect on $J_1$.

On the other hand, the fluctuation of $J_1$ due to SLC directly results in the spin-wave broadening of CrGeTe₃. Therefore, we utilized Monte-Carlo simulations to reproduce the observed magnon spectra (Fig. 4). In the simulation, we calculate several magnon spectra with different $J_1$, and sum them up after multiplying a Gaussian coefficient centered at $J_1 = -2.76$ meV. In this case, the width of the Gaussian coefficient is directly related to the extent of the $J_1$ variations. In order to reproduce the magnon broadening of the optical modes at 3.5 K, a 15% and 25% (in FWHM) variation of $J_1$ is needed, respectively [Fig. 4(a, b, e, f)].

The difference of the broadening effect between Fig. 4(a) and (b) may be attributed to SLCs with different atomic vibrational modes and will have different effect on $J$ fluctuation. Nevertheless, the simulated $J_1$ variation of 0.42~0.71 meV is close to the estimated value of 0.72 meV, suggesting that the observed magnon broadening can be understood as SLC induced by the lattice vibrations. Upon increasing temperature to $T \approx 0.85 T_C = 55$ K, spin waves are softened by ~5% but a 35% variation of $J_1$ is needed to explain the magnon broadening [Fig. 4(c)]. For comparison, the energy width of spin waves along the L direction is resolution limited at 3.5 K [Fig. 1(e)], and only broadens by ~ 5% at 50 K confirming the anisotropic nature of the SLC [Fig. 4(d)]. Since the magnon broadening is observed at 3.5 K, atomic displacement (lattice vibrations) due to thermal fluctuations can be safely ignored and the SLC must be induced by the zero-temperature quantum motion of Cr atoms. Since neutron is a weakly interacting probe, we do not expect neutron scattering itself to affect the populations of magnons and phonons at these temperatures.

In a local moment system with spin S, the total moment sum rule requires $M_0^2 = M^2 + \langle \mathbf{m}^2 \rangle = g^2 S(S+1)\mu_B^2$, where M is the static ordered moment, $\langle \mathbf{m}^2 \rangle$ is the local fluctuating moment, and $g \approx 2$ is the Landé electron spin g factor, to be a temperature independent constant[34,35]. While the ordered moment in a magnetic ordered system can be directly measured via temperature dependence of the magnetic Bragg peak intensity, the local fluctuating moment $\langle \mathbf{m}^2 \rangle$ can be estimated through integrating the magnetic excitations within the first Brillouin zone overall energies. To test if the total sum rule is satisfied in FM CrGeTe₃, we compare the changes in magnetic Bragg peak intensity together with spin waves integrated within the first Brillouin zone. Figure 4(g) shows the temperature evolution of the (1, 1, 0) Bragg peak intensity. The 150 K data (black) has nuclear and incoherent background scattering but without static magnetic scattering. The additional intensity in the low-temperature data comes solely from static ordered moment M, and is proportional to $M^2$. The moment M is expected to reach $gS\mu_B$ at zero temperature. By integrating the elastic magnetic intensity at 3.5 K, we find S = 1.45 consistent with the S = 3/2 picture. At this temperature, the local

fluctuating moment $\langle \mathbf{m}^2 \rangle$ should integrate up to $S \sim 1.5$ to complete the sum rule of $S(S+1)$. Figure 4(h) shows the calculated INS intensity using LSWT with the integration of $(S_{xx} + S_{yy} + S_{zz})$ yielding $S = 1.5$ (blue dashed line). The experimental INS intensity, although preserving the overall shape, is over 50% less than the expectation values of a $S = 3/2$ ferromagnet, suggesting that spin waves in CrGeTe$_3$ do not follow the local moment LSWT.

Upon increasing temperature to 55 K, the ordered moment decreases and yields an effective $S = 1.03$ (Fig. 4(g)). To maintain the sum rule, the local fluctuating moment $\langle \mathbf{m}^2 \rangle$ should integrate up to an effective $S = 2.7$. This intensity increase is mostly due to the Bose factor as well as the additional intensity at the neutron energy gain side (in which spin excitations transfer energy to the incident neutron). The red dashed line in Fig. 4(h) shows the LSWT calculated intensity at 55 K, with the integration of $(S_{xx} + S_{yy} + S_{zz}) = 2.7$. Comparing with experimental data, the observed spin waves show larger intensity reduction at 55 K, thus further suggesting violation of the total moment sum rule.

In summary, we systematically investigate the magnon and phonon spectrum as well as the SLC in CrGeTe$_3$ using INS and DFT calculations. Our results reveal the existence of a strong, highly anisotropic SLC mainly affecting magnons along the in-plane directions. The strong SLC severely shortens the in-plane magnon lifetime even with minimal thermal fluctuation at low temperatures, making CrGeTe$_3$ the first example to have visible effect of SLC caused by quantum zero-point motion of the lattice. This means the spin excitations in CrGeTe$_3$ will dissipate regardless of their topological nature[26]. Therefore, CrGeTe$_3$ will not be suitable for dissipationless spintronics, but can be an ideal candidate for pressure sensitive spintronic devices due to its exceptionally large and anisotropic SLC.

## Methods

**Single-crystal growth and reciprocal space.** High-quality single crystals of CrGeTe$_3$ was made by Ge-Te flux method[36]. The crystals are typically $5 \times 5$ mm$^2$ within the $ab$ plane and a few micron thick along the $c$-axis. Although CrGeTe$_3$ belongs to the rhombohedral $R\text{-}3$ space group, we use a hexagonal lattice with $a = b = 6.86$ Å, $c = 20.42$ Å as shown in Fig. 1(a, b) to describe its crystal structure. In this notation, the momentum transfer $\mathbf{Q} = H\mathbf{a}^* + K\mathbf{b}^* + L\mathbf{c}^*$ is denoted as $(H, K, L)$ in reciprocal lattice units (r.l.u.) [Fig. 1(c)]. The high symmetry points $\Gamma$, $M$, $K$, $A$ in reciprocal space and their, respectively, equivalent points are specified [Fig. 1(c)].

**Neutron scattering.** Neutron diffraction experiments were performed at HB-3 triple-axis spectrometer at High Flux Isotope Reactor, Oak Ridge National Laboratory (ORNL) on a piece of single-crystal sample aligned in the $[H, H, L]$ scattering plane. Temperature dependence of the $(1, 1, 0)$ Bragg peak shows $T_C = 65$ K and negative in-plane thermal expansion below $T_C$, consistent with previous reports (see supplementary information for additional data and analysis)[36]. To further determine the lattice structural distortion and atomic Debye-Waller factors of Cr, Ge, and Te atoms at different temperatures across $T_C$, we performed time-of-flight neutron diffraction experiments on a piece of single-crystal sample at TOPAZ single-crystal diffraction at Spallation Neutron Source (SNS), ORNL[37]. We measured many diffraction peaks at 150, 70 and 5 K, and used the JANA 2006 software[38] to refine the crystallographic parameters. Table 1 shows the diagonal elements of the atomic displacement matrix ($U_{11} = U_{22}$, $U_{33}$), as well as the isotropic displacement $U_{\text{iso}}$. With decreasing temperature, the lattice displacements of the atoms are highly anisotropic favoring vibration along the $c$-axis. Upon cooling from 150 to 5 K, The $U_{33}$ of the magnetic Cr atom only decreases by ~50% while values of $U_{11}$ and $U_{22}$ are suppressed by an order of magnitude. The same anisotropic suppression of the atomic displacement is also observed in the non-magnetic Ge and Te atoms, but not as strong as that in Cr. This suggests the presence of a strong anisotropic magnetoelastic coupling. We also confirmed the negative thermal expansion of the lattice parameter $a$ (see supplementary information for additional data and analysis) as reported previously[27].

In order to map out the spin waves, time-of-flight INS experiments were performed using the SEQUOIA spectrometer at SNS, ORNL[39] and the AMATERAS spectrometer at MLF, J-PARC, Japan[40]. Single crystals of total mass 0.42 gram were co-aligned on an aluminum plate with the help of an X-ray Laue machine. Five different incident energies of $E_i = 50, 37$ meV (SEQUOIA) and $E_i = 15, 4.7, 1.0$ meV (AMATERAS) were used to carry out INS measurements in

the Horace mode[41]. The magnon band top and bottom appears at $\Gamma$, while the Dirac cone appears at the $K$ point, splitting the higher energy optical and lower energy acoustic magnon bands. We utilized the MATLAB Horace and SpinW package[41,42] to reconstruct the neutron structure factor $S(E, \mathbf{Q})$, convolve with instrumental resolution, and integrate the neutron intensity in the momentum space.

**Harmonic density functional theory simulations.** Phonon simulations were performed in the framework of DFT as implemented in the Vienna Ab-initio Simulation Package (VASP 5.4.1)[43–45]. We used $6 \times 6 \times 6$ gamma-centered Monkhorst-Pack electronic $k$-point mesh to integrate over the Brillouin zone for the rhombohedral unit cell with 10 atoms. The plane-wave cut-off energy of 450 eV provided satisfactory degree of convergence (energy difference of ~ 0.1 meV/atom). The convergence criteria for the self-consistent electronic loop was set to $10^{-8}$ eV. The projector-augmented-wave potentials explicitly included six valence electrons for Cr ($4s^2 3d^4$), four for Ge ($3s^2 3p^2$) and six for Te ($4s^2 4p^4$). We performed spin-polarized calculations without spin-orbit coupling, with a magnetic momentum of 2.91 $\mu_B$ on each Cr atom. We used the generalized gradient approximation (GGA) in the Perdew-Burke-Ernzerhof (PBE) parametrization[46,47]. During the relaxation of the unit cell, the lattice parameters and atomic positions were optimized until forces on all atoms were smaller than 1 meV Å$^{-1}$. The resulting lattice parameters were $a = b = c = 7.8364$ Å, and $\alpha = \beta = \gamma = 51.9824°$. Phonon dispersions were calculated in the harmonic approximation, using the finite displacement approach as implemented in Phonopy[48]. The phonon calculations used a $2 \times 2 \times 2$ supercell of the rhombohedral cell containing 80 atoms. The atomic displacement amplitude was 0.01 Å.

**Phonon intensity simulations.** The simulated wave vector-resolved phonon intensity was calculated using the following expression:

$$S(\mathbf{Q}, E) \propto \sum_s \sum_\tau \frac{1}{\omega_s}$$
$$\times \left| \sum_d \frac{b_d^{\text{coh}}}{\sqrt{M_d}} \exp(-W_d) \exp(i\mathbf{Q} \cdot \mathbf{d})(\mathbf{Q} \cdot \mathbf{e}_{ds}) \right|^2 \quad (3)$$
$$\times \left\langle n_s + \frac{1}{2} \pm \frac{1}{2} \right\rangle \delta(\omega \mp \omega_s) \delta(\mathbf{Q} - \mathbf{q} - \tau)$$

where $b_d^{\text{coh}}$ is the coherent neutron scattering length for atom $d$, $\mathbf{Q} = \mathbf{k} - \mathbf{k}'$ is the wave vector transfer, $\mathbf{d}$ the equilibrium position of atom $d$, $\mathbf{e}_{ds}$ the eigenvector of phonon mode s for atom $d$, and $\mathbf{k}'$ and $\mathbf{k}$ are the final and incident wave vector of the scattered particle, $\mathbf{q}$ the phonon wave vector, $\omega_s$ the eigenvalue of the phonon corresponding to the branch index $s$, $\tau$ is the reciprocal lattice vector, $d$ the atom index in the unit cell, $\exp(-2W_d)$ the corresponding DW factor, and $n_s = \left[ \exp\left( \frac{\hbar \omega_s}{k_B T} \right) - 1 \right]^{-1}$ is the Bose-Einstein occupation factor ($E = \hbar \omega_s$). The $+$ and $-$ sign in Eq. (3) correspond to phonon creation and phonon annihilation, respectively. The phonon eigenvalues and eigenvectors in Eq. (3) were obtained by solving dynamical matrix using Phonopy[48].

## Data availability
The data that support the plots within this paper and other findings of this study are available from the corresponding authors upon reasonable request.

## Code availability
The codes used for the DFT calculations of phonon spectra in this study are available from the corresponding authors upon reasonable request.

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

## Acknowledgements

We thank Dr. Seiko Ohira-Kawamura for her help in experiments on AMATERAS and Prof. R. Q. Wu for discussions. The neutron scattering and sample growth work at Rice is supported by U.S. NSF-DMR-2100741 and by the Robert A. Welch Foundation under Grant No. C-1839, respectively (P.D.). The works of JHC was supported by the National Research Foundation (NRF) of Korea (Grant nos. 2020R1A5A1016518 and 2020K1A3A7A09077712). Work at Duke (first-principles modeling) was supported by the U.S. Department of Energy, Office of Science, Basic Energy Sciences, Materials Sciences and Engineering Division, under Award No. DE-SC0019978. A portion of this research used resources at the Spallation Neutron Source, a DOE Office of Science User Facilities operated by the Oak Ridge National Laboratory. The neutron experiment at the Materials and Life Science Experimental Facility of the J-PARC was performed under a user program proposal number 2020A0099.

## Author contributions

P.D., L.C., and J.H.C. conceived the project. L.C. and B.G. grew the single crystals and aligned them using a Laue X-ray diffractometer. The neutron scattering experiments were carried out by L.C., J.H.C., M.B.S., A.I.K., X.P.W., N.M., and P.D. Phonon calculations are carried out by C.M. and O.D. The paper was written by P.D., L.C., C.M., and O.D. and all coauthors made comments on the paper.

## Competing interests

The authors declare no competing interests.
