## [Peer Review File · Nature Communications]

Reviewers' Comments:

Reviewer #1:

Remarks to the Author:

The authors present an inelastic neutron scattering study of the 2D van der Waals ferromagnet CrGeTe₃ complemented by ab-initio lattice dynamical and linear spin wave theory calculations. Furthermore, results from neutron single crystal diffraction are presented. The main point of the report is that magnons dispersing along [HH0] show pronounced broadening effects whereas magnons dispersing along [00L] are resolution limited. Also, the conservation of the number of magnetic excitations seems to be violated in the ferromagnetic state. Following previous discussions on a strong dependence of the Cr-Cr nearest neighbour magnetic interaction on their distance, the authors assign the damping of the in-plane magnons to dynamical spin-lattice coupling arising from the zero-temperature lattice fluctuations in CrGeTe₃.

The recent discovery of ferromagnetic order in single-layers of 2D van der Waals materials such as CrGeTe₃ and CrI₃ triggered large scientific interest because of possible technical applications. But these materials also pose fundamental questions on the stability of magnetic order in two dimensional systems. Hence, the study of these materials with inelastic neutron scattering, in particular, is a topical subject and has been addressed recently for both CrGeTe₃ [Science Advances 7, 7532 (2021)] and CrI₃ [Physical Review B 101, 134418]. The current study goes beyond previous reports in that it provides a quantitative analysis of the life time of the magnons in CrGeTe₃.

However, there are some points because of which I cannot support publication of this report in Nature communications:

- (1) The presented interpretation, i.e., the strong dependence of the Cr-Cr magnetic interaction on the interatomic distance, has been discussed (as correctly pointed out by the authors) in previous literature. Specifically, the spin-lattice coupling in optic phonon modes has been assigned to this mechanism in first principles studies. Also, broad spin waves have been reported. The current manuscript reports the impact of this mechanism on low-energy magnons but does not represent a significant step forward in the understanding of the physics of 2D van der Waals ferromagnets.
- (2) The authors claim to present a combined magnon and phonon study. However, it is much more a magnon study showing also some data for phonons. The topic of spin-lattice coupling would definitely benefit from a more thorough analysis of phonons.
- (3) In my opinion, the issue of the experimental resolution is not addressed adequately regarding its paramount importance for a quantitative analysis of the magnon linewidth. The use of a composite sample requires an even more detailed attention to this subject.

Here are some more comments:

- (a) Why do the authors only discuss c-axis polarized phonons?
When magnons show strong damping in-plane, what is the behaviour of corresponding in-plane polarized phonon modes, which would change the nearest neighbour Cr-Cr distance? The time-of-flight experiments should provide such data not only around (0,0,12) but also for Brillouin zones with similar large absolute wave vectors in the a-b plane, right?
- (b) It would be interesting to see the crossover from the observed low-energy magnon dispersion (Fig. 3d) to that observed for higher energies in a direct comparison. In which q range does it happen?
- (c) It would be instructive, in my opinion, to visualize the anisotropic atomic displacement parameters. The values given in table 1 would also benefit from a graphical presentation.

(d) It is noted that the in-plane data are obtained by averaging over large sections in L, [-5:5] and [-6:6]. This is standard practice to improve statistics in 2D systems – but the magnetic order in CrGeTe₃ is still 3D in a crystal and there is a finite dispersion along the out-of-plane direction (see Fig. 4h). For instance, what is the dispersion along the line $Q = (0.1, 0.1, L)$ or $(0.2, 0.2, L)$? In my opinion, integrating over many Brillouin zones along L affects the linewidths of the data – the q dependence of which is the main result. Please explain why you think this is not so or negligible!

(e) One more question regarding (d): Is the structure factor of the magnon constant as function of L? Why? Otherwise, I would assume that the integration over L includes regions with and without a magnon peak.

(f) Related to the point above: when a mode has an increasingly steep dispersion, effects of finite momentum resolution may change the width of the peak. Please provide details on corresponding considerations?

(g) The magnon dispersion along [0,0,L] is shown in Figures 1d and 4h, but no explanation on the binning is given – what is the binning?

(h) The observed magnon line width is approximated by $E_{FWHM} = \gamma H^2 + E_0$. E_0 “accounts for instrumental energy resolution at different incident neutron energy and L values”. Surely, one also needs to include effects of energy transfer and momentum resolution?!

(i) Please provide the constant Q cuts and approximated peak functions for the results shown in Figure 3c along with analogue data for the out-of-plane dispersion.

(j) All of the above mentioned points regarding resolution effects are complicated by using a composite sample with large mosaic.

Reviewer #2:

Remarks to the Author:

In recent years, magnetic (quasi) 2D materials attracted significant attention due to the possible applications. CrGe(Si)Te₃ is a particularly interesting system due to the strong spin-lattice coupling. In the manuscript, authors presented very detailed study of the nature of spin-lattice coupling in CrGeTe₃ which is crucial for potential application of the material. Furthermore, the same “recipe” can be expanded to the class and would be very interesting to see how spin-lattice coupling is evolving in CrGe(Si)Te₃. The manuscript is well written which allows non-experts to easily read and follow. I only have a couple of remarks:

(1) More details about the samples characterisation should be presented and the possible impact of disorder commented.

(2) The font sizes in figures should be increased since it is very difficult to read in printed version.

Finally, I recommend publishing the manuscript after this are addressed properly.

Reviewer #3:

Remarks to the Author:

I have read through the manuscript “Anisotropic magnon damping by zero-temperature CrGeTe₃” by L. Chen et al. The authors present an interesting and rather systematic study on the magnons and phonons in two-dimensional van der Waals honeycomb lattice ferromagnetic CrGeTe₃ by using inelastic neutron scattering and DFT calculations. The experimental data is of high quality and the article is well written. Their neutron scattering result show anomalous dispersion and broadening in the in-plane spin waves which cannot be explained based on the Heisenberg model and instrumental resolution. The authors found the presence of dynamical spin-

lattice coupling (SLC) arising from the zero-temperature quantum fluctuations of Cr atoms can explain these magnon band anomalies and damping. The authors have indeed produced a good estimate for the microscopic spin Hamiltonian in these systems. Evidence of violation of the total moment sum rule has also been found for the 1st time in CrGeTe₃. This is a well-executed neutron scattering study, that has also been well-analyzed. The interpretation is sound and the theoretical modelling solid. The research on SLC in this system has reference significance in the selection of materials for spintronics.

However, before I can recommend publication of the manuscript, I still got some questions and comments:

Samples that aligned have a mosaic width of about 8.9° (authors mentioned in the supplementary), maybe it is OK for measuring the magnon band dispersion, but it is also large enough to bring a considerable broadening effect in the measured spectrum. Has the broadening effect arise from mosaic width been considered when fitting the broadening in the simulation?

The experimental results in Fig2(a)(b) are integrated results over a large range of L which covers multiple period along [00L] direction, e.g. $L = [-5, 5]$. Although CrGeTe₃ is indeed a 2D vdW layered system, but the magnon band still got a little bit dispersion along [00L] due to the weak inter-layer exchange interactions, as show in Fig1(d). So, the integration process of the TOF data can also introduce extrinsic band broadening in energy artificially. Has the effect also included when doing the simulation?

The above two broadening effect is comparable with the broadening effect caused by the SLC? Or can be neglected?

The key point of the anisotropic magnon damping is that the in-plane exchange constant J_1 is much more sensitive to the atomic displacement than the other exchange constant J_2, J_3, J_c . Since the neutron diffraction refinement is done as well as the temperature dependence of lattice parameters, all the needed parameters including Debye Waller factor are determined, principally the SLC coefficient dJ_1/du and the fluctuation of J_1 can be also estimated according to DFT calculations e.g. in ref[21,22].

Then why not give it a try to directly use these calculated parameters to reproduce the magnon spectrum, and furthermore the authors can compare it with the previous fitted fluctuation of J_1 ("15%" and "35%") to check how good the modified spin Hamiltonian works.

As for checking the total moment sum rule, usually it will be difficult for TOF of single crystal samples to cover the whole FBZ to do the integration . Even for powder samples, the integration of elastic and inelastic part will always smaller than the total moment, e.g. about 5-10%. How much % is the integration part smaller than the total moment in CrGeTe₃? It will be much more intuitive if that value can be given. I also noticed the energy was mapping up to 25 meV in Fig4(j), but in Fig4(f) the energy of magnon optical band can reach about 30meV, does the energy above 25 meV should be included when checking the total moment sum rule? Is there any particular connection between the SLC and the deviation of the total moment sum rule?

Reviewer #1 (Remarks to the Author):

The authors present an inelastic neutron scattering study of the 2D van der Waals ferromagnet CrGeTe₃ complemented by ab-initio lattice dynamical and linear spin wave theory calculations. Furthermore, results from neutron single crystal diffraction are presented. The main point of the report is that magnons dispersing along [HH0] show pronounced broadening effects whereas magnons dispersing along [00L] are resolution limited. Also, the conservation of the number of magnetic excitations seems to be violated in the ferromagnetic state. Following previous discussions on a strong dependence of the Cr-Cr nearest neighbor magnetic interaction on their distance, the authors assign the damping of the in-plane magnons to dynamical spin-lattice coupling arising from the zero-temperature lattice fluctuations in CrGeTe₃.

The recent discovery of ferromagnetic order in single layers of 2D van der Waals materials such as CrGeTe₃ and CrI₃ triggered large scientific interest because of possible technical applications. But these materials also pose fundamental questions on the stability of magnetic order in two dimensional systems. Hence, the study of these materials with inelastic neutron scattering, in particular, is a topical subject and has been addressed recently for both CrGeTe₃ [Science Advances 7, 7532 (2021)] and CrI₃ [Physical Review B 101, 134418]. The current study goes beyond previous reports in that it provides a quantitative analysis of the lifetime of the magnons in CrGeTe₃.

However, there are some points because of which I cannot support publication of this report in Nature communications:

(1) The presented interpretation, i.e., the strong dependence of the Cr-Cr magnetic interaction on the interatomic distance, has been discussed (as correctly pointed out by the authors) in previous literature. Specifically, the spin-lattice coupling in optic phonon modes has been assigned to this mechanism in first principles studies. Also, broad spin waves have been reported. The current manuscript reports the impact of this mechanism on low-energy magnons but does not represent a significant step forward in the understanding of the physics of 2D van der Waals ferromagnets.

We first would like to thank the referee for the constructive comments. Whereas the spin-lattice coupling (SLC) has been reported and/or discussed in previous literatures, the SLC that we report in this manuscript is fundamentally different from those observed in the form of lattice constant contraction or optical phonon hardening. The previous observations of the SLC were on the couplings between long-range magnetic ordering and Cr-Cr bond strength/distances across T_C . Our observation, in contrast, addresses the effect of the anisotropic zero-point quantum fluctuations destabilizing the long-range spin waves at $T \rightarrow 0$. This effect was first predicted theoretically in Ref. [22], for which our work presents the first experimental observation. We would like to stress that what we observe at 4 K is not thermal broadening of magnons, which otherwise should be normal behaviors of long-range ferromagnets at temperatures approaching T_C . Although magnon broadenings at low temperatures were revealed in several van der Waals ferromagnets, Refs. [24, 25, 26, 27, 30], the phenomenon has never been discussed in the context of the zero-point quantum fluctuations.

(2) The authors claim to present a combined magnon and phonon study. However, it is much more a

magnon study showing also some data for phonons. The topic of spin-lattice coupling would definitely benefit from a more thorough analysis of phonons.

While we agree with the referee that more thorough analysis of phonons would be desirable, the referee did not specify what kind of analysis is needed and what kind of new information would be acquired through these additional analysis. We have obtained high-Q phonon data in our time of flight measurements and compare them with theoretical calculations, and found good agreement. In the future, we plan to carry out additional measurements on the system, which is clearly beyond the scope of the present paper.

(3) In my opinion, the issue of the experimental resolution is not addressed adequately regarding its paramount importance for a quantitative analysis of the magnon linewidth. The use of a composite sample requires an even more detailed attention to this subject.

We completely agree with these comments from the referee and note that all calculations included in the submitted manuscript were convoluted by the energy-dependent Gaussian functions to account for instrumental resolutions and the sample mosaic spread. In the revised SI, we include the energy-dependent full-width-half-maximums of the time-of-flight spectrometers which were used in this work. While the momentum resolution of the time-of-flight neutron scattering data is mostly due to the sample mosaic spread, the contribution due to the scattering ellipsoid is typically negligible at low energy transfer and not significant even at high energy transfer [R. Kajimoto et al., AIP Conference Proceedings 1969, 050004 (2018)].

Here are some more comments:

(a) Why do the authors only discuss c-axis polarized phonons?

When magnons show strong damping in-plane, what is the behaviour of corresponding in-plane polarized phonon modes, which would change the nearest neighbour Cr-Cr distance? The time-of-flight experiments should provide such data not only around (0,0,12) but also for Brillouin zones with similar large absolute wave vectors in the a-b plane, right?

The magnon-phonon coupling that we propose in this work does not single out particular phonon vibrations but involves any phonon modes that may potentially perturb Cr-Cr exchanges. For this reason, the acoustic phonons also strongly contribute to the magnon lifetime. From the experimental point of view, the phonons with c-axis polarizations appear with stronger intensities as exemplified by the anisotropic thermal factors. Although easier to observe, the displacements along the c axis do not significantly change the bond distance. Instead, the in-plane displacements directly affect the bond distances, from which we have estimated the subsequent changes in exchange constant to the correct order of magnitude.

(b) It would be interesting to see the crossover from the observed low-energy magnon dispersion (Fig. 3d) to that observed for higher energies in a direct comparison. In which q range does it happen?

We estimate that the crossover occurs around $HH \sim 0.2$ r.l.u.. In the revise SI, we include the new 2D plot of E versus HH^3 revealing the full experimental dispersion and comparisons with calculations. This plot shows that the magnon dispersion calculated using the spin Hamiltonian in the main text accounts for the high-energy data in the range above $E > 10$ meV. Whereas the low-energy part of this

calculated dispersion overlaps closely with $E = 272H^2 + 0.1$, the data below $E < 10$ meV follows the curve $E = 158H^2 + 0.1$ much better with smaller stiffness.

(c) It would be instructive, in my opinion, to visualize the anisotropic atomic displacement parameters. The values given in table 1 would also benefit from a graphical presentation.

This is a good suggestion, and we include an additional figure depicting the three dimensional shape of the thermal ellipsoids.

(d) It is noted that the in-plane data are obtained by averaging over large sections in L, [-5:5] and [-6:6]. This is standard practice to improve statistics in 2D systems – but the magnetic order in CrGeTe3 is still 3D in a crystal and there is a finite dispersion along the out-of-plane direction (see Fig. 4h). For instance, what is the dispersion along the line $Q = (0.1, 0.1, L)$ or $(0.2, 0.2, L)$? In my opinion, integrating over many Brillouin zones along L affects the linewidths of the data – the q dependence of which is the main result. Please explain why you think this is not so or negligible!

We apologize for the confusion arising due to the insufficient explanation of the data and calculations. The referee is correct in that 2D plots in Figs. 2(a-b) and 4(a-h) are integrated over multiple Brillouin zones along the L index. We opted to do so because these plots were intended to show the overall magnon band structure exhibiting topological Dirac gaps and energy broadening at high energy parts. In contrast, the plot in Fig. 3(a) is integrated only over a narrow range of $2.8 < L < 3.2$ to highlight the anomalous low-energy dispersion. Similarly, the plots in Fig. 2 of the SI are also integrated over a comparable range of L. We revised the text to explicitly include this information. We stress that all model calculations in this work included the L-dependent dispersions.

(e) One more question regarding (d): Is the structure factor of the magnon constant as function of L? Why? Otherwise, I would assume that the integration over L includes regions with and without a magnon peak.

Indeed, the magnon structure factor is not a function of L index. It is because the honeycomb plane of CrI3 involves negligible buckling. As a result, two Cr sublattice spins are placed on the same z coordinate causing the ferromagnetic CrI3 effectively to have one spin per unit cell. In other words, the two sublattice spins will have identical phase factors for the spin waves traveling along the c axis. This is the reason why the magnon intensity apparently does not change as a function of L except for monotonic decrease due to magnetic form factor.

(f) Related to the point above: when a mode has an increasingly steep dispersion, effects of finite momentum resolution may change the width of the peak. Please provide details on corresponding considerations?

As we have discussed in the item (3) above, the momentum resolution of time-of-light inelastic spectrometer is negligible particularly as low energies. We certainly have included the effect of the sample mosaic spread in our calculations.

(g) The magnon dispersion along $[0, 0, L]$ is shown in Figures 1d and 4h, but no explanation on the binning is given – what is the binning?

For the $[0, 0, L]$ dispersion, the binning along E is 0.05 meV, and the binning along L is 0.05 r. l. u.

These numbers are now added in the caption of Fig. 1d and 4h.

(h) The observed magnon line width is approximated by $E_{FWHM} = \gamma H^2 + E_0$. E_0 “accounts for instrumental energy resolution at different incident neutron energy and L values”. Surely, one also needs to include effects of energy transfer and momentum resolution?!

As discussed in the item (3) above, we include the energy-dependent instrumental resolution function in the revised SI. The said E_0 corresponds to this function. While the resolution function does not explicitly depend on momentum transfer, the reason why E_0 changes with L is because the magnon energy has a dependence on L.

(i) Please provide the constant Q cuts and approximated peak functions for the results shown in Figure 3c along with analogue data for the out-of-plane dispersion.

We included constant-Q cuts in the revised SI.

(j) All of the above mentioned points regarding resolution effects are complicated by using a composite sample with large mosaic.

Again, we have included the effect of sample mosaic spread in the model calculations.

Reviewer #2 (Remarks to the Author):

In recent years, magnetic (quasi) 2D materials attracted significant attention due to the possible applications. CrGe(Si)Te₃ is a particularly interesting system due to the strong spin-lattice coupling. In the manuscript, authors presented very detailed study of the nature of spin-lattice coupling in CrGeTe₃ which is crucial for potential application of the material. Furthermore, the same "recipe" can be expanded to the class and would be very interesting to see how spin-lattice coupling is evolving in CrGe(Si)Te₃. The manuscript is well written which allows non-experts to easily read and follow. I only have a couple of remarks:

(1) More details about the samples characterisation should be presented and the possible impact of disorder commented.

First of all, we thank the referee for the positive evaluation of our manuscript. In the revised draft of the paper, we added our bulk magnetization data. Since we have carried out single crystal refinement of the neutron diffraction data at different temperatures, we find no evidence of site disorder and this is commented in the revised draft.

(2) The font sizes in figures should be increased since it is very difficult to read in printed version.

We apologize for the inconvenience. Figures will be updated with increased font sizes.

Finally, I recommend publishing the manuscript after this are addressed properly.

Reviewer #3 (Remarks to the Author):

I have read through the manuscript "Anisotropic magnon damping by zero-temperature CrGeTe₃" by L. Chen et al. The authors present an interesting and rather systematic study on the magnons and phonons in two-dimensional van der Waals honeycomb lattice ferromagnetic CrGeTe₃ by using inelastic neutron scattering and DFT calculations. The experimental data is of high quality and the article is well written. Their neutron scattering result show anomalous dispersion and broadening in the in-plane spin waves which cannot be explained based on the Heisenberg model and instrumental resolution. The authors found the presence of dynamical spin-lattice coupling (SLC) arising from the zero-temperature quantum fluctuations of Cr atoms can explain these magnon band anomalies and damping. The authors have indeed produced a good estimate for the microscopic spin Hamiltonian in these systems. Evidence of violation of the total moment sum rule has also been found for the 1st time in CrGeTe₃. This is a well-executed neutron scattering study, that has also been well-analyzed. The interpretation is sound and the theoretical modelling solid. The research on SLC in this system has reference significance in the selection of materials for spintronics.

However, before I can recommend publication of the manuscript, I still got some questions and comments:

We appreciate these comments from the referee that accurately describe the paper and his/her commendation concerning publication.

Samples that aligned have a mosaic width of about 8.9° (authors mentioned in the supplementary), maybe it is OK for measuring the magnon band dispersion, but it is also large enough to bring a considerable broadening effect in the measured spectrum. Has the broadening effect arise from mosaic width been considered when fitting the broadening in the simulation?

There are two mosaic spreads that may result in apparent excitation broadening: in-plane and out-of-plane. While the in-plane mosaic spread is relatively larger, it has little effect on the low-energy part of the magnon because the magnon dispersion is nearly isotropic. Its effect will become noticeable at high energies as the magnon approaches significantly close to the Brillouin zone boundaries. The out-of-plane mosaic spread may also cause broadening as the magnon dispersions are anisotropic between in-plane and out-of-plane directions. To estimate how much broadening can occur in experimental data, we performed Monte-Carlo simulations including three trivial effects: sample mosaic spread in 3D, instrumental resolution and integrations over finite momentum, but excluding the SLC effect. The results, included as new figures in the revised SI, clearly demonstrate that the observed excitation broadening cannot be accounted for only by the non-SLC effect including the sample mosaic spread. In addition, we would like to point out that the magnon excitation of CrI₃ [L. Chen et al., Phys. Rev.

X 8, 041028 (2018)] exhibited significantly smaller broadening although the mosaic spread of the sample was larger (17 degrees in-plane and ~ 8 degrees out-of-plane). This is noted in the revised draft.

The experimental results in Fig2(a)(b) are integrated results over a large range of L which covers multiple period along [00L] direction, e.g. $L = [-5, 5]$. Although CrGeTe₃ is indeed a 2D vdW layered system, but the magnon band still got a little bit dispersion along [00L] due to the weak inter-layer exchange interactions, as show in Fig1(d). So, the integration process of the TOF data can also introduce extrinsic band broadening in energy artificially. Has the effect also included when doing the simulation?

Yes, indeed we calculated the integration over large L ranges for comparison with the experimental data.

The above two broadening effect is comparable with the broadening effect caused by the SLC? Or can be neglected?

The L-dependent dispersion is certainly much larger than the effect of the SLC-induced broadening. However, our claim of the SLC-induced broadening is based on the data shown in Figs. 1(e) and 3(a) [and Fig. 2(a) of the SI], which are integrated over very narrow L ranges where the L-dispersion is negligible. In the revised version, we will explicitly provide the relevant information.

The key point of the anisotropic magnon damping is that the in-plane exchange constant J_1 is much more sensitive to the atomic displacement than the other exchange constant J_2, J_3, J_c . Since the neutron diffraction refinement is done as well as the temperature dependence of lattice parameters, all the needed parameters including Debye Waller factor are determined, principally the SLC coefficient dJ_1/du and the fluctuation of J_1 can be also estimated according to DFT calculations e.g. in ref[21,22].

Then why not give it a try to directly use these calculated parameters to reproduce the magnon spectrum, and furthermore the authors can compare it with the previous fitted fluctuation of J_1 ("15%" and "35%") to check how good the modified spin Hamiltonian works.

We appreciate the idea brought up by the referee, and we have calculated the atomic displacement as a function of temperature using DFT. The details can be found in supplemental information section 3 and Fig. S5 (e). The result shows that although the refinement result of ADP is slightly larger than DFT calculations, they are still in good agreement. Furthermore, the estimation of the J_1 fluctuation of both results agrees well with the observed broadening (15~25% at base, ~35% at 55K).

As for checking the total moment sum rule, usually it will be difficult for TOF of single crystal samples to cover the whole FBZ to do the integration . Even for powder samples, the integration of elastic and inelastic part will always smaller than the total moment, e.g. about 5-10%. How much % is the integration part smaller than the total moment in CrGeTe₃? It will be much more intuitive if that value can be given. I also noticed the energy was mapping up to 25 meV in Fig4(j), but in Fig4(f) the energy of magnon optical band can reach about 30meV, does the energy above 25 meV should be included when checking the total moment sum rule? Is there any particular connection between the SLC and the deviation of the total moment sum rule?

As we have shown in Fig.4 (i, j), The elastic (110) peak intensity is consistent with $S = 3/2$ with only 3% deviation, yet the inelastic intensity is off by more than 50% under the base temperature 3.5K. We will add this to the manuscript.

The integration region in the q-space of the sum rule is specified in the SI sec.4, and a visualization

can be found in fig. S8 in the SI. In the [H K] plane, the sum rule only covers the first Brillouin zone. As we can see in fig.2 (a, b) and fig.4 (b, f), the neutron structure factor is almost zero above 25meV within the first Brillouin zone ($(-1/2, 0) < [H, 0] < (1/2, 0)$, and $(-1/3, -1/3) < [H, H] < (1/3, 1/3)$). Therefore, the inclusion of energies over 25meV will not affect the conclusion that the sum rule is violated.

The main reason we discussed the sum rule is to argue that the neutron excitations in CrGeTe₃ is not 100% magnonic, because otherwise its structure factor will be solely determined by the Cr spin and magnetic form factor, so the total moment sum rule will be preserved because the total quasiparticle number is preserved, and magnons will have infinite lifetime. A deviation from the sum rule suggests that the excitations are hybridized from more than one origin. Given that the magnon-magnon interactions are weak in ferromagnets, the only possibility left is magnon-phonon hybridized excitation.

Reviewers' Comments:

Reviewer #1:

Remarks to the Author:

I thank the authors for the detailed response to the points raised. Overall, I think that now an expert reader can make a realistic assessment of the results and their interpretation. The revised text also puts the relevance of the new results into better perspective to previous work.

Thus, I support the publication of this manuscript after the authors have addressed a few remaining issues, which I list below.

(1) In the introduction, it is stated that "it is important to understand interactions of magnetic excitations with lattice vibrations". Later, the broadening of the spin waves is assigned to the presence of lattice quantum fluctuations ruling out collective lattice vibrations by considering their negligible population at $T = 3.5$ K (bottom of page 6). I agree that this argument is often used and certainly applicable in many cases. Using neutron spectroscopy to study magnetic excitations, however, also creates phonons. Since typical inelastic measurements take several hours, the sample has constantly some finite phonon population at energies well above that corresponding to the sample temperature. These phonons could interact with the magnetic excitations

I would like the authors to include a statement/discussion on this issue. This argument would also make a more detailed phonon study more interesting, since one should be able to observe the interaction also the other way around. However, I would not expect to see it for c-axis polarized lattice vibrations in a layered magnet.

(2) I thank the authors for being more precise on the (H,K,L) binning used to produce the plots. However, for some of them it is still not possible to get detailed information. I can understand that mentioning all the different binning values in a figure caption is bad for readability and might confuse the general readership. But since several vastly different binning ranges are used, I think it is necessary to provide this information. It could be done in the SI, e.g., in a table for all figures together. That way, one would not have to hunt for comments on binning in captions or the text itself. Here some examples, where I did not find the full information:

- a. Figure 1e,f
- b. Figure 2e,f
- c. Figure 4
- d. Figure S3
- e. Figure S4 includes the binning along $[-K,K]$ and $[0,0,L]$ but not for the main direction $[H,H]$
- f. Figure S6

(3) Figure 3c,d: why are there no magnon energies shown deduced from the data in Fig. 1f, i.e., for larger values of H? If one can give a linewidth, certainly, a peak position can also be given. Furthermore, it would help the reader to have the horizontal axes in c and d to have the same length and labels in r.l.u.

(4) Last paragraph of RESULTS: "where E_0 accounts for instrumental energy resolution at different incident neutron energy". The respective instrumental energy resolution at the position of the magnon peak changes with the peak position. Therefore, I do not understand how E_0 can be a constant. Please explain. I thought the resolution was already taken care of before and the values in Figure 3c are the intrinsic broadening.

(5) Please rename figures in the supplemental information as they are referenced in the text, e.g., Fig. S3, etc.

(6) Figure S3c: Thanks for showing the L dependence – what is the binning? It would also be nice to see the L dependence starting at the M point in the data set shown in Fig. 2a, which would show (I expect) that the dispersion of the magnon at high energy is small/negligible.

Reviewer #2:

Remarks to the Author:

After revision, the authors additionally improved the manuscript and addressed all the concerns. It presents valuable addition towards understanding nature of spin-lattice coupling in $\text{CrGe}(\text{Si})\text{Te}_3$ and I recommend it for publication.

Reviewer #3:

None

REVIEWER COMMENTS

Reviewer #1 (Remarks to the Author):

I thank the authors for the detailed response to the points raised. Overall, I think that now an expert reader can make a realistic assessment of the results and their interpretation. The revised text also puts the relevance of the new results into better perspective to previous work.

Thus, I support the publication of this manuscript after the authors have addressed a few remaining issues, which I list below.

(1) In the introduction, it is stated that "it is important to understand interactions of magnetic excitations with lattice vibrations". Later, the broadening of the spin waves is assigned to the presence of lattice quantum fluctuations ruling out collective lattice vibrations by considering their negligible population at $T = 3.5$ K (bottom of page 6). I agree that this argument is often used and certainly applicable in many cases. Using neutron spectroscopy to study magnetic excitations, however, also creates phonons. Since typical inelastic measurements take several hours, the sample has constantly some finite phonon population at energies well above that corresponding to the sample temperature. These phonons could interact with the magnetic excitations

I would like the authors to include a statement/discussion on this issue. This argument would also make a more detailed phonon study more interesting, since one should be able to observe the interaction also the other way around. However, I would not expect to see it for c-axis polarized lattice vibrations in a layered magnet.

Reply: First, we would like to thank the reviewer for the appreciation of our revised work as well as the new constructive comments.

Although neutron scattering certainly excites phonons and can determine phonon dispersion, it is widely accepted that the neutrons will not change phonon populations significantly from thermal equilibrium, which is determined by the sample temperature. Only in extreme cases at mK temperature regime, scattering of neutrons will increase sample temperature and thus increase phonon population. This is because neutrons only weakly interact with sample, and the phonon population is entirely determined by the sample temperature and obeys detailed balance factor. To our knowledge, there is no observable effect of neutron irradiation time on thermal equilibrium of materials in the temperature regime of this experiment.

In our experiments, the ratio between the positive (which we show in the manuscript, corresponding to neutrons losing energy and exciting phonons/magnons) and negative (neutrons gaining energy and de-exciting phonons/magnons) sides of the neutron intensities follows the detailed balance factor. If phonons were created well beyond thermal equilibrium, they would appear anomalously strong in the negative side of the spectrum as well. We do not see such anomalous intensity on the negative side in our data.

To address this issue, we added a sentence to say explicitly we don't expect neutron will affect the thermal dynamics of the sample. The sentence "Since neutron is a weakly interacting probe, we do not expect neutron scattering itself to affect the populations of magnons and phonons at these temperatures." is added near the bottom of page 6 as suggested by the referee.

(2) I thank the authors for being more precise on the (H,K,L) binning used to produce the plots. However, for some of them it is still not possible to get detailed information. I can understand that mentioning all the different binning values in a figure caption is bad for readability and might confuse the general readership. But since several vastly different binning ranges are used, I think it is necessary to provide this information. It could be done in the SI, e.g., in a table for all figures together. That way, one would not have to hunt for comments on binning in captions or the text itself. Here some examples, where I did not find the full information:

- a. Figure 1e,f
- b. Figure 2e,f
- c. Figure 4
- d. Figure S3
- e. Figure S4 includes the binning along $[-K,K]$ and $[0,0,L]$ but not for the main direction $[H,H]$
- f. Figure S6

Reply: This has been properly addressed in table 1 in the SI.

(3) Figure 3c,d: why are there no magnon energies shown deduced from the data in Fig. 1f, i.e., for larger values of H? If one can give a linewidth, certainly, a peak position can also be given. Furthermore, it would help the reader to have the horizontal axes in c and d to have the same length and labels in r.l.u.

Reply: We appreciate these comments. The magnon energy is already displayed in Fig. 1f in the raw data. For dispersion at higher energies, one can use exchange parameters given in main text of the paper. The exchange parameters fitted from the overall spectrum is renormalized and a function of the wavevector transfer. The reviewer can refer to the fig.1 (attached here for convenience) of this paper <https://doi.org/10.1016/j.jmmm.2007.04.010>, where they discussed SLC renormalization of 2D ferromagnetic square lattices. In the " $\alpha_1=0.3$ " case where α_1 is proportional to dJ/du , if one fits the spectrum with a sinusoidal function (dashed red line which is scaled from the $E_{m0}(k)$ line in the figure), one will seem to find most deduced spin excitation energy in the middle of the magnon band instead of the band top, while in fact, the band top has the largest energy difference than its non-renormalized counterpart $E_{m0}(k)$.

We have changed the horizontal axes in fig.3(c, d) accordingly.

(4) Last paragraph of RESULTS: “where E_0 accounts for instrumental energy resolution at different incident neutron energy”. The respective instrumental energy resolution at the position of the magnon peak changes with the peak position. Therefore, I do not understand how E_0 can be a constant. Please explain. I thought the resolution was already taken care of before and the values in Figure 3c are the intrinsic broadening.

Reply: The reviewer is correct about this issue, and we are sorry for not clarifying it in the text. In the last revision, the data points shown in fig. 3(c) are already subtracted by the calculated instrumental resolution, and the 0.2meV FWHM at $[H H] = 0$ is from the integration in the Q-space and sample mosaic. We have properly addressed it in the main text.

(5) Please rename figures in the supplemental information as they are referenced in the text, e.g., Fig. S3, etc.

Reply: This has been properly addressed.

(6) Figure S3c: Thanks for showing the L dependence – what is the binning? It would also be nice to see the L dependence starting at the M point in the data set shown in Fig. 2a, which would show (I expect) that the dispersion of the magnon at high energy is small/negligible.

Reply: Here the binning for $[0 0 L]$ is 0.05 (r. l. u.), and the binning for E is 0.2meV. The L dispersion at $[H H] = 0.5$ as well as $[-K K] = 0.5$ is shown here. With considerable SLC at this Q, it is hard to determine any dispersive modes for the L dispersion, therefore we are not able to determine J_{C1} and J_{C2} in the way we had successfully done in CrI_3 (PhysRevX.11.031047).

Reviewer #2 (Remarks to the Author):

After revision, the authors additionally improved the manuscript and addressed all the concerns. It presents valuable addition towards understanding nature of spin-lattice coupling in $CrGe(Si)Te_3$ and I recommend it for publication.

Reply: We would like to thank the reviewer for the appreciation and recognition of our revised work.

Reviewers' Comments:

Reviewer #1:

Remarks to the Author:

The authors have addressed all questions. I recommend publication.